# High-Order Attention Models for Visual Question Answering

**Idan Schwartz**
Department of Computer Science
Technion
idansc@cs.technion.ac.il

**Alexander G. Schwing**
Department of Electrical and Computer Engineering
University of Illinois at Urbana-Champaign
aschwing@illinois.edu

**Tamir Hazan**
Department of Industrial Engineering & Management
Technion
tamir.hazan@gmail.com

## Abstract

The quest for algorithms that enable cognitive abilities is an important part of machine learning. A common trait in many recently investigated cognitive-like tasks is that they take into account different data modalities, such as visual and textual input. In this paper we propose a novel and generally applicable form of attention mechanism that learns high-order correlations between various data modalities. We show that high-order correlations effectively direct the appropriate attention to the relevant elements in the different data modalities that are required to solve the joint task. We demonstrate the effectiveness of our high-order attention mechanism on the task of visual question answering (VQA), where we achieve state-of-the-art performance on the standard VQA dataset.

## 1  Introduction

The quest for algorithms which enable cognitive abilities is an important part of machine learning and appears in many facets, *e.g.*, in visual question answering tasks [6], image captioning [26], visual question generation [18, 10] and machine comprehension [8]. A common trait in these recent cognitive-like tasks is that they take into account different data modalities, for example, visual and textual data.

To address these tasks, recently, attention mechanisms have emerged as a powerful common theme, which provides not only some form of interpretability if applied to deep net models, but also often improves performance [8]. The latter effect is attributed to more expressive yet concise forms of the various data modalities. Present day attention mechanisms, like for example [15, 26], are however often lacking in two main aspects. First, the systems generally extract abstract representations of data in an ad-hoc and entangled manner. Second, present day attention mechanisms are often geared towards a specific form of input and therefore hand-crafted for a particular task.

To address both issues, we propose a novel and generally applicable form of attention mechanism that learns high-order correlations between various data modalities. For example, second order correlations can model interactions between two data modalities, *e.g.*, an image and a question, and more generally, $k-$th order correlations can model interactions between $k$ modalities. Learning these correlations effectively directs the appropriate attention to the relevant elements in the different data modalities that are required to solve the joint task.

We demonstrate the effectiveness of our novel attention mechanism on the task of visual question answering (VQA), where we achieve state-of-the-art performance on the VQA dataset [2]. Some

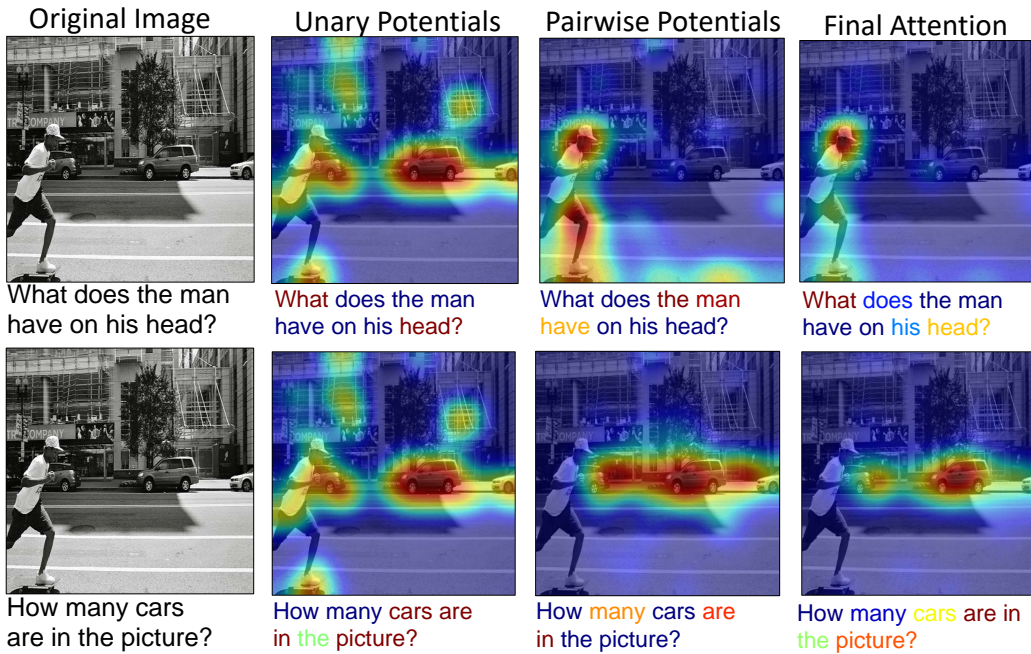

| Original Image | Unary Potentials | Pairwise Potentials | Final Attention |
|---|---|---|---|

What does the man have on his head?

How many cars are in the picture?

Figure 1: Results of our multi-modal attention for one image and two different questions (1st column). The unary image attention is identical by construction. The pairwise potentials differ for both questions and images since both modalities are taken into account (3rd column). The final attention is illustrated in the 4th column.

of our results are visualized in Fig. 1, where we show how the visual attention correlates with the textual attention.

We begin by reviewing the related work. We subsequently provide details of our proposed technique, focusing on the high-order nature of our attention models. We then conclude by presenting the application of our high-order attention mechanism to VQA and compare it to the state-of-the-art.

## 2 Related work

Attention mechanisms have been investigated for both image and textual data. In the following we review mechanisms for both.

**Image attention mechanisms:** Over the past few years, single image embeddings extracted from a deep net (*e.g.*, [17, 16]) have been extended to a variety of image attention modules, when considering VQA. For example, a textual long short term memory net (LSTM) may be augmented with a spatial attention [29]. Similarly, Andreas *et al.* [1] employ a language parser together with a series of neural net modules, one of which attends to regions in an image. The language parser suggests which neural net module to use. Stacking of attention units was also investigated by Yang *et al.* [27]. Their stacked attention network predicts the answer successively. Dynamic memory network modules which capture contextual information from neighboring image regions has been considered by Xiong *et al.* [24]. Shih *et al.* [23] use object proposals and and rank regions according to relevance. The multi-hop attention scheme of Xu *et al.* [25] was proposed to extract fine-grained details. A joint attention mechanism was discussed by Lu *et al.* [15] and Fukui *et al.* [7] suggest an efficient outer product mechanism to combine visual representation and text representation before applying attention over the combined representation. Additionally, they suggested the use of glimpses. Very recently, Kazemi *et al.* [11] showed a similar approach using concatenation instead of outer product. Importantly, all of these approaches model attention as a single network. The fact that multiple modalities are involved is often not considered explicitly which contrasts the aforementioned approaches from the technique we present.

Very recently Kim *et al.* [14] presented a technique that also interprets attention as a multi-variate probabilistic model, to incorporate structural dependencies into the deep net. Other recent techniques are work by Nam *et al.* [19] on dual attention mechanisms and work by Kim *et al.* [13] on bilinear

models. In contrast to the latter two models our approach is easy to extend to any number of data modalities.

**Textual attention mechanisms:** We also want to provide a brief review of textual attention. To address some of the challenges, *e.g.*, long sentences, faced by translation models, Hermann *et al.* [8] proposed RNNSearch. To address the challenges which arise by fixing the latent dimension of neural nets processing text data, Bahdanau *et al.* [3] first encode a document and a query via a bidirectional LSTM which are then used to compute attentions. This mechanism was later refined in [22] where a word based technique reasons about sentence representations. Joint attention between two CNN hierarchies is discussed by Yin *et al.* [28].

Among all those attention mechanisms, relevant to our approach is work by Lu *et al.* [15] and the approach presented by Xu *et al.* [25]. Both discuss attention mechanisms which operate jointly over two modalities. Xu *et al.* [25] use pairwise interactions in the form of a similarity matrix, but ignore the attentions on individual data modalities. Lu *et al.* [15] suggest an alternating model, that directly combines the features of the modalities before attending. Additionally, they suggested a parallel model which uses a similarity matrix to map features for one modality to the other. It is hard to extend this approach to more than two modalities. In contrast, our model develops a probabilistic model, based on high order potentials and performs mean-field inference to obtain marginal probabilities. This permits trivial extension of the model to any number of modalities.

Additionally, Jabri *et al.* [9] propose a model where answers are also used as inputs. Their approach questions the need of attention mechanisms and develops an alternative solution based on binary classification. In contrast, our approach captures high-order attention correlations, which we found to improve performance significantly.

Overall, while there is early work that propose a combination of language and image attention for VQA, *e.g.*, [15, 25, 12], attention mechanism with several potentials haven't been discussed in detail yet. In the following we present our approach for joint attention over any number of modalities.

# 3 Higher order attention models

Attention modules are a crucial component for present day decision making systems. Particularly when taking into account more and more data of different modalities, attention mechanisms are able to provide insights into the inner workings of the oftentimes abstract and automatically extracted representations of our systems.

An example of such a system that captured a lot of research efforts in recent years is Visual Question Answering (VQA). Considering VQA as an example, we immediately note its dependence on two or even three different data modalities, the visual input $V$, the question $Q$ and the answer $A$, which get processed simultaneously. More formally, we let

$$V \in \mathbb{R}^{n_v \times d}, \quad Q \in \mathbb{R}^{n_q \times d}, \quad A \in \mathbb{R}^{n_a \times d}$$

denote a representation for the visual input, the question and the answer respectively. Hereby, $n_v$, $n_q$ and $n_a$ are the number of pixels, the number of words in the question, and the number of possible answers. We use $d$ to denote the dimensionality of the data. For simplicity of the exposition we assume $d$ to be identical across all data modalities.

Due to this dependence on multiple data modalities, present day decision making systems can be decomposed into three major parts: (i) the data embedding; (ii) attention mechanisms; and (iii) the decision making. For a state-of-the-art VQA system such as the one we developed here, those three parts are immediately apparent when considering the high-level system architecture outlined in Fig. 2.

## 3.1 Data embedding

Attention modules deliver to the decision making component a succinct representation of the relevant data modalities. As such, their performance depends on how we represent the data modalities themselves. Oftentimes, an attention module tends to use expressive yet concise data embedding algorithms to better capture their correlations and consequently to improve the decision making performance. For example, data embeddings based on convolutional deep nets which constitute the state-of-the-art in many visual recognition and scene understanding tasks. Language embeddings heavily rely on LSTM which are able to capture context in sequential data, such as words, phrases and sentences. We give a detailed account to our data embedding architectures for VQA in Sec. 4.1.

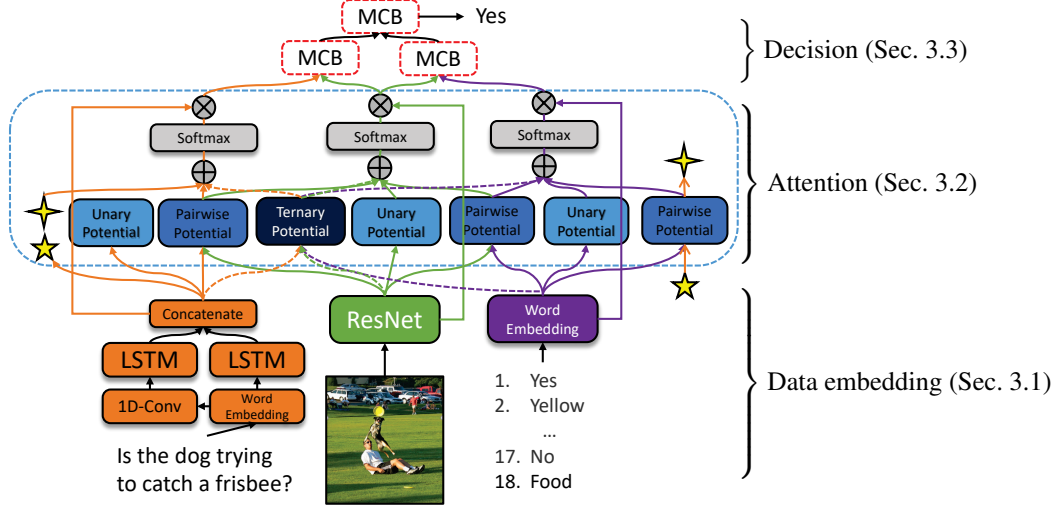

Figure 2: Our state-of-the-art VQA system

## 3.2 Attention

As apparent from the aforementioned description, attention is the crucial component connecting data embeddings with decision making modules.

Subsequently we denote attention over the $n_q$ words in the question via $P_Q(i_q)$, where $i_q \in \{1, \ldots, n_q\}$ is the word index. Similarly, attention over the image is referred to via $P_V(i_v)$, where $i_v \in \{1, \ldots, n_v\}$, and attention over the possible answers are denoted $P_A(i_a)$, where $i_a \in \{1, \ldots, n_a\}$.

We consider the attention mechanism as a probability model, with each attention mechanism computing "potentials." First, unary potentials $\theta_V, \theta_Q, \theta_A$ denote the importance of each feature (*e.g.*, question word representations, multiple choice answers representations, and image patch features) for the VQA task. Second, pairwise potentials, $\theta_{V,Q}, \theta_{V,A}, \theta_{Q,A}$ express correlations between two modalities. Last, third-order potential, $\theta_{V,Q,A}$ captures dependencies between the three modalities.

To obtain marginal probabilities $P_Q$, $P_V$ and $P_A$ from potentials, our model performs mean-field inference. We combine the unary potential, the marginalized pairwise potential and the marginalized third order potential linearly including a bias term:

$$
\begin{aligned}
P_V(i_v) &= \mathrm{smax}(\alpha_1\theta_V(i_v)+\alpha_2\theta_{V,Q}(i_v)+\alpha_3\theta_{A,V}(i_v)+\alpha_4\theta_{V,Q,A}(i_v)+\alpha_5), \\
P_Q(i_q) &= \mathrm{smax}(\beta_1\theta_Q(i_q)+\beta_2\theta_{V,Q}(i_q)+\beta_3\theta_{A,Q}(i_q)+\beta_4\theta_{V,Q,A}(i_q)+\beta_5), \\
P_A(i_a) &= \mathrm{smax}(\gamma_1\theta_A(i_a)+\gamma_2\theta_{A,V}(i_a)+\gamma_3\theta_{A,Q}(i_a)+\gamma_4\theta_{V,Q.A}(i_a)+\gamma_5).
\end{aligned}
\tag{1}
$$

Hereby $\alpha_i$, $\beta_i$, and $\gamma_i$ are learnable parameters and $\mathrm{smax}(\cdot)$ refers to the soft-max operation over $i_v \in \{1, \ldots, n_v\}$, $i_q \in \{1, \ldots, n_q\}$ and $i_a \in \{1, \ldots, n_a\}$ respectively. The soft-max converts the combined potentials to probability distributions, which corresponds to a single mean-field iteration. Such a linear combination of potentials provides extra flexibility for the model, since it can learn the reliability of the potential from the data. For instance, we observe that question attention relies more on the unary question potential and on pairwise question and answer potentials. In contrast, the image attention relies more on the pairwise question and image potential.

Given the aforementioned probabilities $P_V$, $P_Q$, and $P_A$, the attended image, question and answer vectors are denoted by $a_V \in \mathbb{R}^d$, $a_Q \in \mathbb{R}^d$ and $a_A \in \mathbb{R}^d$. The attended modalities are calculated as the weighted sum of the image features $V = [v_1, \ldots, v_{n_v}]^T \in \mathbb{R}^{n_v \times d}$, the question features $Q = [q_1, \ldots, q_{n_q}]^T \in \mathbb{R}^{n_q \times d}$, and the answer features $A = [a_1, \ldots, a_{n_a}]^T \in \mathbb{R}^{n_a \times d}$, *i.e.*,

$$
a_V = \sum_{i_v=1}^{n_v} P_V(i_v)v_{i_v}, \quad a_Q = \sum_{i_q=1}^{n_q} P_Q(i_q)q_{i_q}, \quad \text{and} \quad a_V = \sum_{i_a=1}^{n_a} P_A(i_a)a_{i_a}.
$$

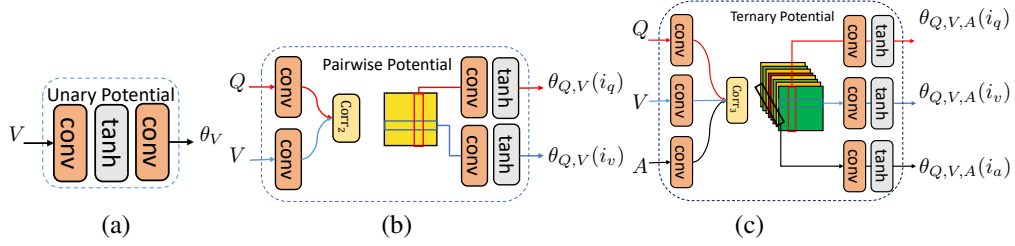

Figure 3: Illustration of our $k-$order attention. (a) unary attention module (e.g., visual). (b) pairwise attention module (e.g., visual and question) marginalized over its two data modalities. (c) ternary attention module (e.g., visual, question and answer) marginalized over its three data modalities..

The attended modalities, which effectively focus on the data relevant for the task, are passed to a classifier for decision making, *e.g.*, the ones discussed in Sec. 3.3. In the following we now describe the attention mechanisms for unary, pairwise and ternary potentials in more detail.

### 3.2.1 Unary potentials

We illustrate the unary attention schematically in Fig. 3 (a). The input to the unary attention module is a data representation, *i.e.*, either the visual representation $V$, the question representation $Q$, or the answer representation $A$. Using those representations, we obtain the 'unary potentials' $\theta_V$, $\theta_Q$ and $\theta_A$ using a convolution operation with kernel size $1 \times 1$ over the data representation as an additional embedding step, followed by a non-linearity ($\tanh$ in our case), followed by another convolution operation with kernel size $1 \times 1$ to reduce embedding dimensionality. Since convolutions with kernel size $1 \times 1$ are identical to matrix multiplies we formally obtain the unary potentials via

$$\theta_V(i_v) = \tanh(VW_{v_2})W_{v_1}, \quad \theta_Q(i_q) = \tanh(QW_{q_2})W_{q_1}, \quad \theta_A(i_a) = \tanh(AW_{a_2})W_{a_1}.$$

where $W_{v_1}, W_{q_1}, W_{a_1} \in \mathbb{R}^{d \times 1}$, and $W_{v_2}, W_{q_2}, W_{a_2} \in \mathbb{R}^{d \times d}$ are trainable parameters.

### 3.2.2 Pairwise potentials

Besides the mentioned mechanisms to generate unary potentials, we specifically aim at taking advantage of pairwise attention modules, which are able to capture the correlation between the representation of different modalities. Our approach is illustrated in Fig. 3 (b). We use a similarity matrix between image and question modalities $C_2 = QW_q(VW_v)^\top$. Alternatively, the $(i, j)$-th entry is the correlation (inner-product) of the $i$-th column of $QW_q$ and the $j$-th column of $VW_v$:

$$(C_2)_{i,j} = \text{corr}_2((QW_q)_{:,i}, (VW_v)_{:,j}), \qquad \text{corr}_2(q, v) = \sum_{l=1}^{d} q_l v_l.$$

where $W_q, W_v \in \mathbb{R}^{d \times d}$ are trainable parameters. We consider $(C_2)_{i,j}$ as a pairwise potential that represents the correlation of the $i$-th word in a question and the $j$-th patch in an image. Therefore, to retrieve the attention for a specific word, we convolve the matrix along the visual dimension using a $1 \times 1$ dimensional kernel. Specifically,

$$\theta_{V,Q}(i_q) = \tanh\left(\sum_{i_v=1}^{n_v} w_{i_v}(C_2)_{i_v,i_q}\right), \quad \text{and} \quad \theta_{V,Q}(i_v) = \tanh\left(\sum_{i_q=1}^{n_q} w_{i_q}(C_2)_{i_v,i_q}\right).$$

Similarly, we obtain $\theta_{A,V}$ and $\theta_{A,Q}$, which we omit due to space limitations. These potentials are used to compute the attention probabilities as defined in Eq. (1).

### 3.2.3 Ternary Potentials

To capture the dependencies between all three modalities, we consider their high-order correlations.

$$(C_3)_{i,j,k} = \text{corr}_3((QW_q)_{:,i}, (VW_v)_{:,j}, (AW_a)_{:,k}), \qquad \text{corr}_3(q, v, a) = \sum_{l=1}^{d} q_l v_l a_l.$$

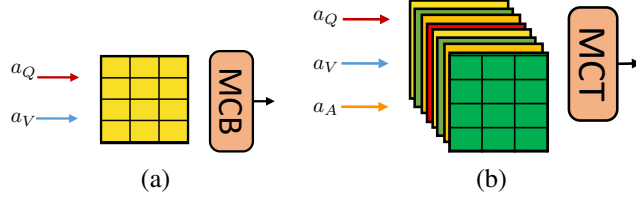

<div align="center">(a)                         (b)</div>

Figure 4: Illustration of correlation units used for decision making. (a) MCB unit approximately sample from outer product space of two attention vectors, (b) MCT unit approximately sample from outer product space of three attention vectors.

Where $W_q, W_v, W_a \in \mathbb{R}^{d \times d}$ are trainable parameters. Similarly to the pairwise potentials, we use the $C_3$ tensor to obtain correlated attention for each modality:

$$\theta_{V,Q,A}(i_q) = \tanh\left(\sum_{i_v=1}^{n_v} \sum_{i_a=1}^{n_a} w_{i_v,i_a}(C_3)_{i_q,i_v,i_a}\right), \; \theta_{V,Q,A}(i_v) = \tanh\left(\sum_{i_q=1}^{n_q} \sum_{i_a=1}^{n_a} w_{i_q,i_a}(C_3)_{i_q,i_v,i_a}\right),$$

$$\text{and} \quad \theta_{V,Q,A}(i_a) = \tanh\left(\sum_{i_v=1}^{n_v} \sum_{i_q=1}^{n_q} w_{i_q,i_a}(C_3)_{i_q,i_v,i_a}\right).$$

These potentials are used to compute the attention probabilities as defined in Eq. (1).

### 3.3 Decision making

The decision making component receives as input the attended modalities and predicts the desired output. Each attended modality is a vector that consists of the relevant data for making the decision. While the decision making component can consider the modalities independently, the nature of the task usually requires to take into account correlations between the attended modalities. The correlation of a set of attended modalities are represented by the outer product of their respective vectors, *e.g.*, the correlation of two attended modalities is represented by a matrix and the correlation of $k$-attended modalities is represented by a $k$-dimensional tensor.

Ideally, the attended modalities and their high-order correlation tensors are fed into a deep net which produces the final decision. The number of parameters in such a network grows exponentially in the number of modalities, as seen in Fig. 4. To overcome this computational bottleneck, we follow the tensor sketch algorithm of Pham and Pagh [21], which was recently applied to attention models by Fukui *et al.* [7] via Multimodal Compact Bilinear Pooling (MCB) in the pairwise setting or Multimodal Compact Trilinear Pooling (MCT), an extension of MCB that pools data from three modalities. The tensor sketch algorithm enables us to reduce the dimension of any rank-one tensor while referring to it implicitly. It relies on the count sketch technique [4] that randomly embeds an attended vector $a \in \mathbb{R}^{d_1}$ into another Euclidean space $\Psi(a) \in \mathbb{R}^{d_2}$. The tensor sketch algorithm then projects the rank-one tensor $\otimes_{i=1}^{k} a_i$ which consists of attention correlations of order $k$ using the convolution $\Psi(\otimes_{i=1}^{k} a_i) = *_{i=1}^{k} \Psi(a_i)$. For example, for two attention modalities, the correlation matrix $a_1 a_2^\top = a_1 \otimes a_2$ is randomly projected to $\mathbb{R}^{d_2}$ by the convolution $\Psi(a_1 \otimes a_2) = \Psi(a_1) * \Psi(a_2)$. The attended modalities $\Psi(a_i)$ and their high-order correlations $\Psi(\otimes_{i=1}^{k} a_i)$ are fed into a fully connected neural net to complete decision making.

## 4 Visual question answering

In the following we evaluate our approach qualitatively and quantitatively. Before doing so we describe the data embeddings.

### 4.1 Data embedding

The attention module requires the question representation $Q \in \mathbb{R}^{n_q \times d}$, the image representation $V \in \mathbb{R}^{n_v \times d}$, and the answer representation $A \in \mathbb{R}^{n_a \times d}$, which are computed as follows.

**Image embedding:** To embed the image, we use pre-trained convolutional deep nets (*i.e.*, VGG-19, ResNet). We extract the last layer before the fully connected units. Its dimension in the VGG net case is $512 \times 14 \times 14$ and the dimension in the ResNet case is $2048 \times 14 \times 14$. Hence we obtain

Table 1: Comparison of results on the Multiple-Choice VQA dataset for a variety of methods. We observe the combination of all three unary, pairwise and ternary potentials to yield the best result.

| | test-dev | | | | test-std |
|---|---|---|---|---|---|
| Method | Y/N | Num | Other | All | All |
| Naive Bayes [15] | 79.7 | 40.1 | 57.9 | 64.9 | - |
| HieCoAtt (ResNet) [15] | 79.7 | 40.0 | 59.8 | 65.8 | 66.1 |
| RAU (ResNet) [20] | 81.9 | 41.1 | 61.5 | 67.7 | 67.3 |
| MCB (ResNet) [7] | - | - | - | 68.6 | - |
| DAN (VGG) [19] | - | - | - | 67.0 | - |
| DAN (ResNet) [19] | - | - | - | 69.1 | 69.0 |
| MLB (ResNet) [13] | - | - | - | - | 68.9 |
| 2-Modalities: Unary+Pairwis (ResNet) | 80.9 | 36.0 | 61.6 | 66.7 | - |
| 3-Modalities: Unary+Pairwise (ResNet) | **82.0** | 42.7 | 63.3 | 68.7 | 68.7 |
| 3-Modalities: Unary + Pairwise + Ternary (VGG) | 81.2 | 42.7 | 62.3 | 67.9 | - |
| 3-Modalities: Unary + Pairwise + Ternary (ResNet) | 81.6 | **43.3** | **64.8** | **69.4** | **69.3** |

$n_v = 196$ and we embed both the 196 VGG-19 or ResNet features into a $d = 512$ dimensional space to obtain the image representation $V$.

**Question embedding:** To obtain a question representation, $Q \in \mathcal{R}^{n_q \times d}$, we first map a 1-hot encoding of each word in the question into a $d$-dimensional embedding space using a linear transformation plus corresponding bias terms. To obtain a richer representation that accounts for neighboring words, we use a 1-dimensional temporal convolution with filter of size 3. While a combination of multiple sized filters is suggested in the literature [15], we didn't find any benefit from using such an approach. Subsequently, to capture long-term dependencies, we used a Long Short Term Memory (LSTM) layer. To reduce overfitting caused by the LSTM units, we used two LSTM layers with $d/2$ hidden dimension, one uses as input the word embedding representation, and the other one operates on the 1D conv layer output. Their output is then concatenated to obtain $Q$. We also note that $n_q$ is a constant hyperparameter, *i.e.*, questions with more than $n_q$ words are cut, while questions with less words are zero-padded.

**Answer embedding:** To embed the possible answers we use a regular word embedding. The vocabulary is specified by taking only the most frequent answers in the training set. Answers that are not included in the top answers are embedded to the same vector. Answers containing multiple words are embedded as n-grams to a single vector. We assume there is no real dependency between the answers, therefore there is no need of using additional 1D conv, or LSTM layers.

## 4.2 Decision making

For our VQA example we investigate two techniques to combine vectors from three modalities. First, the attended feature representation for each modality, *i.e.*, $a_V$, $a_A$ and $a_Q$, are combined using an MCT unit. Each feature element is of the form $((a_V)_i \cdot (a_Q)_j \cdot (a_A)_k)$. While this first solution is most general, in some cases like VQA, our experiments show that it is better to use our second approach, a 2-layer MCB unit combination. This permits greater expressiveness as we employ features of the form $((a_V)_i \cdot (a_Q)_j \cdot (a_Q)_k \cdot (a_A)_t)$ therefore also allowing image features to interact with themselves. Note that in terms of parameters both approaches are identical as neither MCB nor MCT are parametric modules.

Beyond MCB, we tested several other techniques that were suggested in the literature, including element-wise multiplication, element-wise addition and concatenation [13, 15, 11], optionally followed by another hidden fully connected layer. The tensor sketching units consistently performed best.

## 4.3 Results

**Experimental setup:** We use the RMSProp optimizer with a base learning rate of $4e^{-4}$ and $\alpha = 0.99$ as well as $\epsilon = 1e^{-8}$. The batch size is set to 300. The dimension $d$ of all hidden layers is set to 512. The MCB unit feature dimension was set to $d = 8192$. We apply dropout with a rate of 0.5 after the word embeddings, the LSTM layer, and the first conv layer in the unary potential units. Additionally, for the last fully connected layer we use a dropout rate of 0.3. We use the top 3000 most frequent

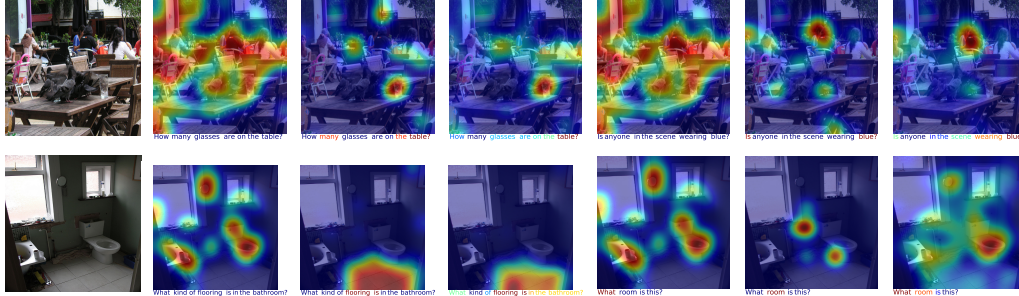

Figure 5: For each image (1st column) we show the attention generated for two different questions in columns 2-4 and columns 5-7 respectively. The attentions are ordered as unary attention, pairwise attention and combined attention for both the image and the question. We observe the combined attention to significantly depend on the question.

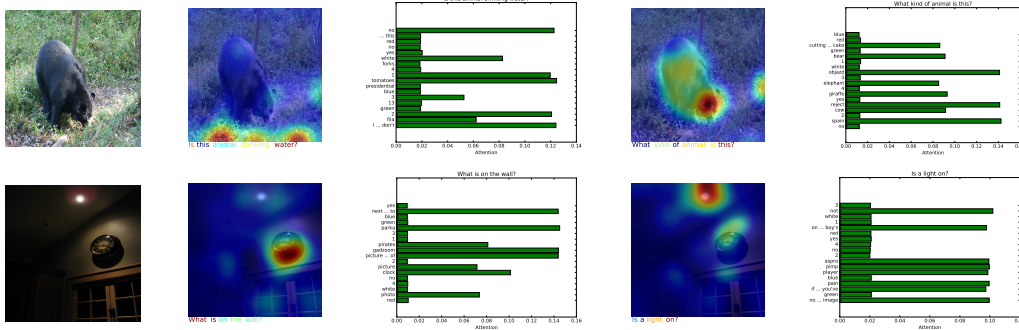

Figure 6: The attention generated for two different questions over three modalities. We find the attention over multiple choice answers to emphasis the unusual answers.

answers as possible outputs, which covers 91% of all answers in the train set. We implemented our models using the Torch framework[1] [5].

As a comparison for our attention mechanism we use the approach of Lu *et al.* [15] and the technique of Fukui *et al.* [7]. Their methods are based on a hierarchical attention mechanism and multi-modal compact bilinear (MCB) pooling. In contrast to their approach we demonstrate a relatively simple technique based on a probabilistic intuition grounded on potentials. For comparative reasons only, the visualized attention is based on two modalities: image and question.

We evaluate our attention modules on the VQA real-image test-dev and test-std datasets [2]. The dataset consists of $123,287$ training images and $81,434$ test set images. Each image comes with 3 questions along with 18 multiple choice answers.

**Quantitative evaluation:** We first evaluate the overall performance of our model and compare it to a variety of baselines. Tab. 1 shows the performance of our model and the baselines on the test-dev and the test-standard datasets for multiple choice (MC) questions. To obtain multiple choice results we follow common practice and use the highest scoring answer among the provided ones. Our approach (Fig. 2) for the multiple choice answering task achieved the reported result after 180,000 iterations, which requires about 40 hours of training on the 'train+val' dataset using a TitanX GPU. Despite the fact that our model has only 40 million parameters, while techniques like [7] use over 70 million parameters, we observe state-of-the-art behavior. Additionally, we employ a 2-modality model having a similar experimental setup. We observe a significant improvement for our 3-modality model, which shows the importance of high-order attention models. Due to the fact that we use a lower embedding dimension of 512 (similar to [15]) compared to 2048 of existing 2-modality models [13, 7], the 2-modality model achieves inferior performance. We believe that higher embedding dimension and proper tuning can improve our 2-modality starting point.

Additionally, we compared our proposed decision units. MCT, which is a generic extension of MCB for 3-modalities, and 2-layers MCB which has greater expressiveness (Sec. 4.2). Evaluating on the 'val' dataset while training on the 'train' part using the VGG features, the MCT setup yields 63.82%

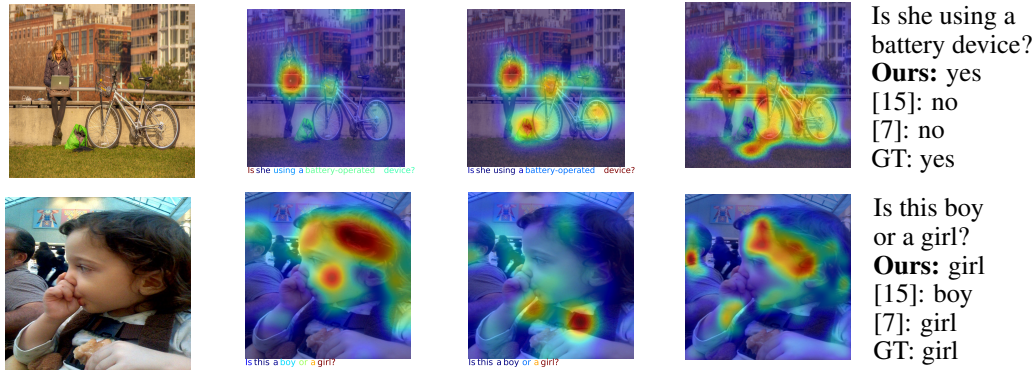

Is she using a
battery device?
**Ours:** yes
[15]: no
[7]: no
GT: yes

Is this boy
or a girl?
**Ours:** girl
[15]: boy
[7]: girl
GT: girl

Figure 7: Comparison of our attention results (2nd column) with attention provided by [15] (3rd column) and [7] (4th column). The fourth column provides the question and the answer of the different techniques.

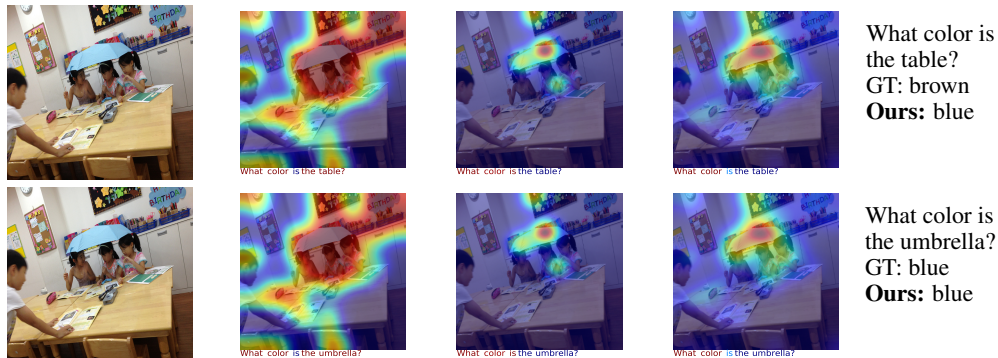

What color is
the table?
GT: brown
**Ours:** blue

What color is
the umbrella?
GT: blue
**Ours:** blue

Figure 8: Failure cases: Unary, pairwise and combined attention of our approach. Our system focuses on the colorful umbrella as opposed to the table in the first row.

where 2-layer MCB yields 64.57%. We also tested a different ordering of the input to the 2-modality MCB and found them to yield inferior results.

**Qualitative evaluation:** Next, we evaluate our technique qualitatively. In Fig. 5 we illustrate the unary, pairwise and combined attention of our approach based on the two modality architecture, without the multiple choice as input. For each image we show multiple questions. We observe the unary attention usually attends to strong features of the image, while pairwise potentials emphasize areas that correlate with question words. Importantly, the combined result is dependent on the provided question. For instance, in the first row we observe for the question "How many glasses are on the table?," that the pairwise potential reacts to the image area depicting the glass. In contrast, for the question "Is anyone in the scene wearing blue?" the pairwise potentials reacts to the guy with the blue shirt. In Fig. 6, we illustrate the attention for our 3-modality model. We find the attention over multiple choice answers to favor the more unusual results.

In Fig. 7, we compare the final attention obtained from our approach to the results obtained with techniques discussed in [15] and [7]. We observe that our approach attends to reasonable pixel and question locations. For example, considering the first row in Fig. 7, the question refers to the battery operated device. Compared to existing approaches, our technique attends to the laptop, which seems to help in choosing the correct answer. In the second row, the question wonders "Is this a boy or a girl?". Both of the correct answers were produced when the attention focuses on the hair.

In Fig. 8, we illustrate a failure case, where the attention of our approach is identical, despite two different input questions. Our system focuses on the colorful umbrella as opposed to the object queried for in the question.

## 5 Conclusion

In this paper we investigated a series of techniques to design attention for multimodal input data. Beyond demonstrating state-of-the-art performance using relatively simple models, we hope that this work inspires researchers to work in this direction.

**Acknowledgments:** This research was supported in part by The Israel Science Foundation (grant No. 948/15). This material is based upon work supported in part by the National Science Foundation under Grant No. 1718221. We thank Nvidia for providing GPUs used in this research.

## Footnotes

[1] https://github.com/idansc/HighOrderAtten

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
