[Supplementary Material]

# Supplementary Material: High-Order Attention Models for Visual Question Answering

## 1 Qualitative results

In Fig. 1 we show additional attention heat maps. For the picture provided in the first column we demonstrate the difference in the obtained attention for two different questions in columns 2-4 and columns 5-7 respectively. Intermediate attention results for the unary and pairwise modalities are provided in columns 2 & 3, and columns 5 & 6 for the two different questions. Column 4 and column 7 depict the final attention result of our mechanism. The unary attention usually attends to strong features of the image. The pairwise potential is created from the correlation of image features with the textual feature. While in many cases the pairwise potential is accurate, it is clear that the combined attention is more accurate in most cases.

In Fig. 4 we compare results of different approaches that use attention for the VQA task. These additional images provide insights regarding the importance of attention to generate the correct answer. In the following we discuss the results illustrated in the left column more carefully, while noting that the results demonstrated in the right column are similar. Left column, first row, the approach of [2] answer "red," while the answer is obviously black. The reason for the answers could be explained by the attention modules. Left column, second row, we observe that attention not focused on the skiers results in a wrong answer for one of the approaches, while ours predicts the correct answer.

We want to also further explain on the process of generating these attention maps. For comparable reasons we used two modalities architecture, which described in Fig. 2. Lu *et al*. [2] generates attention maps for each textual representation, word, phrase, and question. In this analysis, the attention maps are taken from the question (last) representation, because they indicate the most meaningful attention in this model. Fukui *et al*. [1] uses two attention maps, also known as glimpses. As in the demo of [1][1], we are showing the first attention map for their model, because it is visually more plausible.

## 2 Ternary Potentials Implementation

We further describe our implementations of $C_3$ tensor. In a sense $C_3$ is constructed by multiplication of three metrics. Support for high order operations is very limited in deep learning architectures. Our solution is based on existing building blocks, using a dynamic neural networks that depends on the spatial size of the input metrics. First, we split the question tensor over the spatial dimension, Assuming $n_q < n_a < n_v$. For each spatial location $q_i \in \mathbb{R}^d$ we perform a element wise multiplication with each spatial location of $A$, we then perform a matrix multiplication operation with $V$ the result equals to $(C_3)_{1,:} \in \mathbb{R}^{1 \times n_v \times n_a}$. Last, we concatenate the $n_q$ outputs on the first dimension to achieve $C_3 \in \mathbb{R}^{n_q \times n_v \times n_a}$. This process is illustrated in Fig. 3.

## Footnotes

[1]http://demo.berkeleyvision.org

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

Figure 4: Comparison of our attention results (1st column) with attention provided by [2] (2nd column) and [1] (3rd column). The fourth column provides the question and the answer of the different techniques.