[Reviews · NeurIPS 2017]

Reviewer 1



General Impression: Overall I think the proposed method is interesting. The results are quite good and the attention maps seem to illustrate these gains might be due to improved attention mechanisms and not simply increased model capacity. I found the writing to be a bit weak and sometimes a bit confusing, though I imagine given more time the authors could improve the submission appropriately. Strengths: - Interesting model with good results. - Thorough qualitative and quantitative experiments. - I'm fairly impressed by the shift between unary/pairwise attentions and the final attention. Though I would have liked to see marginalized trinary attention maps somewhere as well. Did I miss these in the text or supplement? Weaknesses: - As I said above, I found the writing / presentation a bit jumbled at times. - The novelty here feels a bit limited. Undoubtedly the architecture is more complex than and outperforms the MCB for VQA model [7], but much of this added complexity is simply repeating the intuition of [7] at higher (trinary) and lower (unary) orders. I don't think this is a huge problem, but I would suggest the authors clarify these contributions (and any I may have missed). - I don't think the probabilistic connection is drawn very well. It doesn't seem to be made formally enough to take it as anything more than motivational which is fine, but I would suggest the authors either cement this connection more formally or adjust the language to clarify. - Figure 2 is at an odd level of abstraction where it is not detailed enough to understand the network's functionality but also not abstract enough to easily capture the outline of the approach. I would suggest trying to simplify this figure to emphasize the unary/pairwise/trinary potential generation more clearly. - Figure 3 is never referenced unless I missed it. Some things I'm curious about: - What values were learned for the linear coefficients for combining the marginalized potentials in equations (1)? It would be interesting if different modalities took advantage of different potential orders. - I find it interesting that the 2-Modalities Unary+Pairwise model under-performs MCB [7] despite such a similar architecture. I was disappointed that there was not much discussion about this in the text. Any intuition into this result? Is it related to swap to the MCB / MCT decision computation modules? - The discussion of using sequential MCB vs a single MCT layers for the decision head was quite interesting, but no results were shown. Could the authors speak a bit about what was observed?

Reviewer 2



The paper presents a novel method to model attention on the image and the question for the task of Visual Question Answering (VQA). The main novelty lies in proposing a generic formulation of attention for multiple modalities – a combination of unary, pairwise and ternary potentials for 3 modalities (image, question and answer) in VQA. The proposed approach is evaluated on the VQA dataset and outperforms existing models when ternary potentials are used. Strengths: 1. A generic formulation of attention for multimodal problems that is not specific to a particular task is useful because especially if it can be shown to generalize to more than one task. 2. The ablation study showing the performance without ternary potential is useful to understand how much ternary potential helps. 3. The paper is clearly written. Weaknesses: 1. The approach mentions attention over 3 modalities – image, question and answer. However, it is not clear what attention over answers mean because most of the answers are single words and even if they are multiword, they are treated as single word. The paper does not present any visualizations for attention over answers. So, I would like the authors to clarify this. 2. From the results in table 1, it seems that the main improvement in the proposed model is coming from the ternary potential. Without the ternary potential, the proposed model is not outperforming the existing models for 2 modalities setup (except HieCoAtt). So, I would like the authors to throw light into this. 3. Since ternary potential seems to be the main factor in the performance improvement of the proposed model, I would like the authors to compare the proposed model with existing models where answers are also used as inputs such as Revisiting Visual Question Answering Baselines (Jabri et al., ECCV16). 4. The paper lacks any discussion on failure cases of the proposed model. It would be insightful to look into the failure modes so that future research can be guided accordingly. 5. Other errors/typos: a. L38: mechanism -> mechanisms b. L237 mentions that the evaluation is on validation set. However Table 1 reports numbers on the test-dev and test-std sets? Post-rebuttal comments: Although the authors' response to the concern of "Proposed model not outperforming existing models for 2 modalities" does not sound satisfactory to me due to lack of quantitative evidence, I would like to recommend acceptance because of the generic attention framework for multiple modalities being proposed in the paper and quantitative results of 3-modality attention outperforming SOTA. The quantitative evaluation of the proposed model's attention maps against human attention maps (reported in the rebuttal) also looks good and suggests that the attention maps are more correlation with human maps' than existing models. Although, we don't know what this correlation value is for SOTA models such as MCB, I think it is still significantly better than that for HieCoAtt. I have a question about one of the responses from the authors -- > Authors' response -- “MCB vs. MCT”: MCT is a generic extension of MCB for n-modalities. Specifically for VQA the 3-MCT setup yields 68.4% on test-dev where 2-layer MCB yields 69.4%. We tested other combinations of more than 2-modalities MCB and found them to yield inferior results. Are the numbers swapped here? 69.4% should be for 3-MCT, right? Also, the MCB overall performance in table 1 is 68.6%. So, not sure which number the authors are referring to when they report 68.4%.